# Prioritizing Age-Friendly Domains for Transforming a Mid-Sized American City

**DOI:** 10.3390/ijerph17239103

**Published:** 2020-12-06

**Authors:** Anthony A. Sterns, Harvey L. Sterns, Ann Walter

**Affiliations:** 1iRxReminder LLC, 1768 E. 25th St., Cleveland, OH 44114, USA; 2Business Administration, School of Professional Studies, City University of New York, New York, NY 10001, USA; 3Management Information Systems, College of Business Administration, Kent State University, Kent, OH 44242, USA; 4Institute for Life-Span Development and Gerontology, Professor Emeritus, Department of Psychology, The University of Akron, Akron, OH 44325, USA; sternsh@uakron.edu; 5Department of Family and Community Medicine, Northeast Ohio Medical University (NEOMED), 4209 St. Rt. 44, P.O. Box 95, Rootstown, OH 44272, USA; 6Consortium of Eastern Ohio Master of Public Health Program, Cleveland State University, 2121 Euclid Ave, Cleveland, OH 44115, USA; a.m.walter@vikes.csuohio.edu; 7Medical Mutual of Ohio, 2060 E 9th Street, Cleveland, OH 44115, USA

**Keywords:** survey, questionnaire, age-friendly, age-friendly cities, older people, age-friendliness, use of technology

## Abstract

In May 2019, the city of Akron in the state of Ohio was admitted into AARP’s network of age friendly cities and communities. Akron has a long history of aging services initiative that date back to the 1970s. To provide direction for future aging initiatives, an assessment of Akron’s current state was conducted in early 2020. A survey designed to capture information on the eight Age-friendly domains was designed and mailed to 3000 randomized individuals in Akron’s ten political wards. A total of 656 individuals responded and returned the survey. Akron is rated good to excellent by older Akronites; people want to stay in their neighborhood and in their home. Most Akronites like and use their neighborhood parks, find their streets well-lit, and feel safe walking in their neighborhood. Most respondents rated transportation in Akron as good to very good, but they found sidewalks good to poor. There is a high level of access to social and educational activities and a substantial opportunity to include more people. About two-thirds of respondents participate in faith-based activities, volunteer, and participate in city-sponsored events. Loneliness is not or rarely a problem for three quarters of respondents. Around 56.5% of respondents indicated they disagree they are disconnected from the community. There is high level of access to the Internet and public WiFi in Akron and a substantial opportunity to include more people. Overall, Akron has benefitted from its historical efforts and has the opportunity to impact on more older adults as the older population grows.

## 1. Introduction

Beginning in the early 1970s, Akron showed strong leadership and commitment to implementing policies and creating services that benefit older people. The Akron Metropolitan Housing Authority had already developed special housing for older adults. In 1974, the City of Akron created the Senior Citizen Commission to the Mayor and City Council. In addition, United Way took responsibility to oversee the development of the Area Agency on Aging with funding from the Older Americans Act. These initiatives included establishing a planning process and funding for a three-county area, in which Akron was the largest city with a population approaching 300,000 at that time.

This led to the establishment of services for older adults using existing service providers with other providers added later. This included an information and referral agency, meals-on-wheels and congregate dining, geriatric clinics by the health department, city sponsored senior recreational centers, senior on-demand transportation services, and additional senior cost-supplemented housing. Further, a multi-purpose senior center was established as a cooperative effort between the Akron Metropolitan Housing Authority and the University of Akron with funding from the Area Agency on Aging, United Way and Summit County Welfare Department. The university also launched an AARP Institute of Lifelong Learning and free university-level credit education programming. Another agency created senior job training and job search support. The County of Summit created a response that led to the formation of a cooperative committee that then led to a Robert Wood Johnson Grant to develop a centralized computer allowing for coordinated assessment and services across over 30 agencies.

These services expanded and were maintained in spite of being constrained with a weakening regional economy and the decline in manufacturing for which Akron had become famous as the “rubber capital of the world.” Over time, political priorities and a population shrinking toward 200,000 persons led to a loss of the aggressive earlier support. Akron was not in a county that supported an aging services levy and over time United Way and foundations allocated less to aging services. The City of Akron for a number of years did not have an active Senior Citizen Commission. However, in recent years, the Akron Community Foundation has made aging a priority and has funded services planning research and programming.

In 2016, Mayor Daniel Horrigan was elected and supported the reactivation of the Senior Citizen Commission. As the commission assessed needs and wants of older adults, a series of community listening events was conducted. During 2017, there was discussion regarding the possibility of Akron becoming part of the age friendly cities initiative. With encouragement from Ohio AARP, the idea of formally applying was carefully considered. Support from the Director of the Institute for Life-Span Development and Gerontology, who also was chair of the Commission on Aging as well as the Dean of Arts and Sciences and later Interim President of the University of Akron, greatly facilitated moving forward. Direction Home Akron Canton Area Agency on Aging took major leadership responsibility. In February 2019, the application to join the AARP Network of Age Friendly Communities was submitted by Mayor Horrigan.

To focus on aging initiatives and blend them into the general initiatives to enhance the livability of the city of Akron, an effort to guide the next era of development was organized around the AARP network of Age Friendly Cities and Communities [1,2,3]. The first step in the process was the establishment of a core committee made up of the members of the Commission. The partners in this process include the City of Akron, Direction Home, Akron Canton Area Agency on Aging and Disabilities, the University of Akron, and the City of Akron Senior Citizens Commission to the Mayor and City Council. Members of the Commission on Aging make up the core committee for the Age Friendly Akron initiative. A broader membership of agencies and individuals, the Advisory Committee, was invited to advise the core committee. The development of the assessment involved this group. To guide policy and planning, an assessment of the current state of the City of Akron was planned.

## 2. Materials and Methods 

The study was executed as a randomized cross-sectional study of older residents of the City of Akron. The survey was assembled and mailed through a fulfillment service to ensure the respondents anonymity was maintained. The survey was reviewed and the procedures approved by the University of Akron Institutional Review Board responsible for ethical treatment of human subjects research. Data entry and analysis were conducted by the research team.

### 2.1. Survey Participants

There were 3000 individuals over the age of 50 randomly selected from all 10 wards in the City of Akron. Surveys were mailed along with prepaid return envelopes. A total of 656 (21.9%) individuals responded and returned the survey. This was comparable to the response rates of Cleveland (*n* = 283, 28.3% return rate) and Columbus (*n* = 346, 23.1% return rate). [4,5]

Respondents were predominately female (62.7%), Caucasian (72.4%), and spoke English (87.8%). The age of respondents ranged 50–95 with an average age of 69.3 (s.d. 10) years of age. About two-fifths (40.1%) were married, a quarter divorced (23.9%), one fifth widowed (21.8%), and the remainder never married (14.2%). Respondents most often lived in a home they owned (76%), while others lived in a condominium (7%), a rented apartment (6%), a rented house (5%), a family member’s house (2%), subsidized housing (2%), or institutionalized care or community living (2%).

### 2.2. Instrument

The committee utilized the framework established in the Global Age-friendly Cities Guide by the WHO [1], as well as the conceptual framework and required questions prescribed by the AARP Age-friendly cities and communities’ guidebook [6]. The Age Friendly Akron survey looks at the eight proscribed domains of the Age-friendly cities and communities framework: (1) housing and neighborhoods; (2) outdoor spaces and buildings; (3) transportation and walkability; (4) arts, entertainment, and leisure; (5) respect and social inclusion; (6) civic participation and employment; (7) communication and information; and (8) health and wellness. Based on a gerontechnological focus within the committee, additional emphases on technology and health services availability were incorporated into the survey instrument. The importance of additional questions on access and adoption of technology were recently reported by Marston and van Hoof [7].

The Age-friendly Akron survey instrument is shared in Appendix A. The instrument was designed to provide a description of the state of the respondents in each of the eight domains. The core questions drew from the required guidelines of the program survey template [8] as well as drawing on other nearby cities in Ohio including Columbus, the State’s capital in the center of the state; Cleveland, just north of Akron on the shores of Lake Erie; and Cincinnati, a city in the south of the state along the Ohio River and the border of the State of Kentucky [4,5,9]. 

Section 1 of the survey consisted of 12 questions on housing. Some questions had components that consisted of multiple aspects of living situations to consider and on which to report. The overall desire to live in their home, neighborhood, and in the City of Akron were assessed using five-point Likert scale questions. 

Section 2 consisted of six multipart questions on outdoor spaces and accessibility. Questions asked respondents to describe the state of city infrastructure including evaluation of street lighting, sidewalk maintenance, walkability, and access to buildings and offices.

Section 3 of the survey consisted of nine questions focused on aspects of transportation and access to various kinds of alternative transportation modalities including walking. Several questions asked respondents to indicate all the transportation modalities they utilize, and the list included various on-demand services including SCAT (on demand paratransit), taxis, Uber/Lyft, and others. Several questions asked about weather’s impact on transportation. The City of Akron experiences four distinct seasons; it is hot in the summer and has snow in the winter.

Section 4 of the survey consisted of six multipart questions on arts, leisure, and educational opportunities available to Akron residents. Most questions focused on usage of major facilities (museums, theaters, outdoor venues, and sports facilities) as well as various festival and educational opportunities available throughout the year.

Section 5 of the survey consisted of five questions on respect and social inclusion. These questions asked respondents to indicate the kinds and frequency of interactions they have and to rate the quality of those interactions. One multipart question asked for a rating of the perceived voice older persons have in the community. 

Section 6 of the survey consisted of seven questions on civic participation and employment. Questions asked respondents about their employment status, participation in childcare and volunteering, and quantifying access to these opportunities. A question asking about experiences with agism related to employment is also in this section.

Section 7 of the survey consisted of seven questions on access to information and questions about access to and use of communication technologies that include telephony and Internet services. Questions in the section also assessed confidence in knowing how to obtain information on various services and awareness of specific programs designed to support the distribution of information and connect older persons with available services.

Section 8 of the survey consisted of 17 questions on health and wellness. The opening question asked the respondent to rate their overall health. Questions asked about access to grocery stores, medical and pharmacy services, and other health related services such as dentistry. Other questions focused on the use of and frequency of acute challenges related to health, food availability, medical services, and mental health. Several questions focused on loneliness and relationships with others in the respondent’s family and in the community. Several questions asked about the use of home modifications required to maintain independence.

A final section of the survey included nine demographic questions including gender, age, income, and marital status.

## 3. Results

As a first step in the analysis, the internal reliability of the domain-specific questions was examined. Questions with ordinal or quantitative responses in each domain were isolated and a Cronbach’s α was calculated for each. The results indicate moderate to high internal consistency within each of the domains (see Table 1). These results, in addition to the consistent approach across the other cities within the WHO and AARP Age-friendly communities provided confidence to examine the individual domains. For a complete list of the questions analyzed, see Appendix B, Table A1.

### 3.1. Housing and Neighborhoods

Respondents rate the city of Akron positively with 88.7% reporting a good or better rating and 34.0% rating it as excellent or very good. Considering the respondents in terms of their average income, there are more favorable views in areas (mailing zip codes) with higher incomes. In the three lower income areas, unfavorable ratings are nearly twice as high (13% vs. 25%), but still a minority (see Figure 1 and Table 2).

Respondents indicated they most strongly want to remain in their homes and feel slightly less strongly about remaining in their neighborhoods and in the City of Akron. These feelings are stronger for the oldest respondents. (See Figure 2).

Several issues in the survey related to housing noted concerns either by absence (e.g., “don’t have access”) or uncertainty (e.g., “not sure”). These issues included access to affordable housing (29.4% not sure/15.9% no), pedestrian crossing timings (15.9% not sure/24.2% no), access to snow removal services (19.3% not sure/22.7% no), access to lawn services (19.0% not sure/25.5% no), well maintained sidewalks (5.9% not sure/48.8% no), and access to a reliable handyman (23.3% not sure/30.1% no). There were no geographic differences in the acceptability of sidewalks and concerns were citywide. 

### 3.2. Outdoor Spaces and Buildings

Respondents indicated that public parks were extremely (43.0%) or very important (34.7%), and 74.0% of respondents indicated public parks in the neighborhood were good, very good, or excellent. About 9.8% indicated that parks in their neighborhood were poor, and 15.0% indicated they did not have them. Accessibility of outdoor spaces and buildings is summarized in Figure 3.

Access to park benches (86.2%), a park with accessible trails (67.0%), pathways for bikes and people (70.1%), and parks that are maintained in the winter (79.0%) were all generally viewed as available for a majority of respondents. 

Buildings were also viewed favorably with accessible front doors (81.4%), having automatic door openers (53.4%), and large enough restrooms (46.6%), all or most of the time.

### 3.3. Transportation and Walkability

Respondents indicated an overwhelmingly positive view of public transit in the city of Akron with more than half rating the transit system good (57.3%), very good (23.4%), or excellent (6.8%). Regardless of age, driving themselves was the most frequent mode (85.4%) and being driven the next most frequent mode (8.2%). There was a three-fold increase in those respondents reporting being driven for those 50–75 (5.3%) versus those 76–95 (16.5%) years of age (see Figure 4). As driving is the major mode of transit, it is positive that streets signs are perceived as legible (85.7%). Awareness of driver refresher courses, however, is low at only 13.5%.

Akron has both a public fixed route bus system and a dedicated on-demand bus service known as SCAT. Both services are utilized more by those 75 and under and by a very small percentage of older adults (5.2% combined).

The perceptions about public transit are overwhelmingly positive, although there is a substantial number of respondents indicating uncertainty or no opinion. Rating on access to transport for those with disabilities (61.3% yes/3.4% no/35.3% not sure) and access to reliable transit (71.5% yes/7.8% no/20.6% not sure) reflect this trend. Concerns are higher with respect to perceptions of lighting at public transit stops (34.6% all or most/65.4% some, few, or none), public transit stops having seating (25.5% all or most/74.5% some, few, or none), and public transit stops having shelters from the weather (21.5% all or most/79.6% some, few, or none).

### 3.4. Arts, Entertainment and Leisure

Akron has historically had a rich cultural environment with the University of Akron as its center with a strong art, dance, and theater program, as well as the Akron Symphony, professional and amateur dance companies, active music scene, numerous live theaters and outdoor concert venues, and a well-respected Museum of Art with a focus on modern works as well as an arts district and monthly art walk. Akron is also home to a downtown baseball stadium for the minor league baseball team as well as stadiums for the University of Akron football, baseball, basketball, and track and field teams. Crisscrossing Akron are the historical locks and canals of the Ohio and Erie Canals and the tow path trail, which connects with walking and bike paths that connect parks and greenspaces throughout the city and along the Cuyahoga River and Summit Lake. Akron is also home to the Akron Zoological Park, Stan Hywett Hall and Gardens, several historical cemeteries, and the Akron Toy Museum as well as many city-sponsored and neighborhood art, cultural, and music festivals throughout the year. 

Respondents indicated they have access to social activities (70.3%), educational events (62.7%), and public events (78.9%) (see Figure 5). 

Questions about participation gauge usage rather than perceptions of access. Respondents were asked how often they participate in events with one quarter (26.6%) participating every other week or more, less than a fifth participating monthly (19.0%), and the remaining majority (54.5%) participating less than monthly (29.6%) or never (25.0%). However, there was an indication that more frequent participation was desired with a third of individuals (32.4%) wanting every other week or more, about a third (30%) wanting monthly event participation, and the remaining quarter interested in less than monthly participation (15.4%), or no participation at all (11.9%).

Respondents were asked if they participate in museums and the zoo. About half (50.2%) indicated yes, while about a sixth (17.3%) indicated they do not but would like too, nearly a quarter (23.8%) indicated they do not but had in the past, and the remainder do not participate and have no interest in doing so (8.6%). A similar pattern was found with live theater, with slightly more participation with city-sponsored events and slightly less participation in sporting events.

Questions were also asked about other activity interests. About half of respondents indicated participation in faith-based activities (50.8% yes/9.1% no but would like too), volunteer activities (34.7% yes/23.3% no, but would like too), and family gatherings (74.4% yes/7.3% no, but would like too).

Leisure activities also include continuing education and physical recreation participation. These appear to offer opportunities for capturing great interest from the community. Only 14.0% of respondents indicated they currently participate in continuing education opportunities, while 34.9% of respondents said they did not participate but would like too. With regard to physical recreation, a larger group (33.4%) indicated they do participate and a similar size group (34.4%) said they do not participate but would like too.

### 3.5. Respect and Social Inclusion

The degree of voice that older persons perceive in the community is an important measure of the respect they feel. Their wellbeing is also connected to the amount of social interaction they experience. These are the focus of the questions related to respect and social inclusion.

#### 3.5.1. Community Interactions

Most (70.7%) older adults interact with friends and family on a daily basis and another 7.1% interact at least monthly. Only 3.1% report highly infrequent contact or have no friends and family. A majority of survey respondents report engaging with other age groups either daily (39.8%), weekly (25.6%), or every other week (8.7%). About one fifth engage with other age groups once per month (9.5%) or less than monthly (8.4%), with 8.4% indicating that they never do so.

About one third (32.0%) of respondents indicated that opinions of older people are valued, while half (50.3%) were not sure, and 17.7% indicated they did not think opinions were valued. Similarly, when respondents were asked if older people were respected by the community: 5.9% strongly agreed, 26.4% agreed, 44.8% were not sure, 20.2% disagreed, and 2.8% strongly disagreed. When respondents were asked if they feel disconnected from the community, a majority strongly disagreed (13.2%) or disagreed (43.3%) with the statement. One quarter (25.1%) of respondents were not sure if they felt disconnected, with 15.4% indicating they agree with feeling disconnected and 3.0% strongly agreeing (see Figure 6).

#### 3.5.2. Purpose and Loneliness

Most respondents strongly agreed or agreed that they feel they have a purpose (52.7%), one third (33.2%) reported not being sure, and 14.1% disagreed or strongly disagreed. Feelings of loneliness are never (44.9%) or rarely (31.1%) experienced by most older adult respondents. However, feelings of loneliness are reported sometimes by about one fifth (21.3%) of respondents. Of most concern are the respondents who report loneliness is experienced most (1.1%) or all of the time (1.6%). Spearman’s rho shows a statistically significant correlation between feelings of purpose within a community and feelings of loneliness (r_s_[594]= −2.14, *p* < 0.001). This correlation is significant but small (see Figure 7).

### 3.6. Civic Participation and Employment

About one fifth of respondents reported working full time (19.2%), with 4.8% working part-time, and 1.6% reported themselves as self-employed. When those reporting retired and working (8.2%) are added in, this is about one third of respondents. This is in line with the percentage of 50–67-year-olds who are below the standard retirement age to receive social security. Of those working, 11.0% indicated enjoying working, 9.0% indicated they could not afford to retire, 4.3% were working to maintain healthcare coverage, and 6.7% said they were not yet of retirement age.

There are a groups of respondents who reported themselves as unemployed and seeking work (1.6%), retired and seeking work (6.3%), and underemployed and seeking more work (0.16%). This would be about 50% higher than the unemployment rate for the City of Akron at the time of survey (4.0%) [10]. Some respondents reported providing unpaid childcare (8.2%) and unpaid eldercare (6.6%). Most respondents indicated they had adequate access to volunteer options but less indicated this for job opportunities, see Figure 8.

The majority of respondents (55.7%) reported being retired and not looking for work. Three quarters (73.3%) of respondents indicated they chose to retire, while the remainder (26.7%) reported they did not choose to retire.

### 3.7. Communication and Information

Finding information that is needed and being proactive about seeking information are reported by a large majority of respondents. Less than 5.7% of respondents indicated they were rarely or never able to do so and less than 8.1% were rarely or never proactive about doing so. When asked specifically about assistance with housing, 12.5% indicated they were always able to and 34.0% were able to find assistance most of the time. More concerning was that respondents reported they found assistance with housing sometimes (24.1%), rarely (14.0%), or never (15.3%). Similarly, when asked specifically about knowing where to go for assistance with healthcare, 25.2% of respondents always knew where to go and 38.6% knew where to go most of the time. However, 21.1% only knew where to get assistance with healthcare sometimes, 8.1% rarely, and 9.0% never.

For computers, the older the respondent, the less likely they were to have a computer with WiFi, see Figure 9. Respondents indicated most still have a landline, but a greater number have computers, a smartphone, and WiFi at home (see Figure A2 in Appendix C).

### 3.8. Health and Wellness

The Health and wellness portion of the survey assessed not only respondents’ overall health and mental health, but also other factors that contribute to health outcomes such as food security, healthcare affordability, access to healthcare services and providers, and health insurance. Overall, 69.8% of respondents rated their overall health as very good or good, 24.2% responded with fair health, and 6.1% rated their health poor or very poor.

Additional insight into health can be gained by examining location; access to food, pharmacies, and fitness centers, and affordability. A decreasing number of respondents with very good and good health, and an increasing number of those with fair, poor, or very poor health is observed when zip codes are ordered by highest income to lowest income, as seen in Figure 10a. When asked, on average, how often respondents participate in physical activity, the majority of those who reported very good overall health exercise every day to several times per week (73.7%). A gradual decline is observed between frequency of exercise as overall health rating declines as well (see Figure A3 and Figure A4 in Appendix C).

Access to a full-service grocery, convenience store, and pharmacy in Akron was indicated by 87.1% or more of respondents in all three categories. The need for more options was highlighted in regard to Healthcare facilities and Urgent care centers (31.8% and 32.7% of respondents, respectively).

Affordability of medications, dentures, glasses/contacts, and hearing aids is considered vital to accessing these items and thereby maintaining low risk of other health issues related to medication adherence, falls, depression, nursing home stays, and dependence on family caregivers [11,12]. Generally, health insurance plans available to older adults including the federally funded Medicare plans for those 65 years and above or disabled do not cover dental, vision, and hearing; these items are often paid for out of pocket.

Seventy-eight percent or more of respondents indicated they can afford regular medications and glasses/contacts always or most of the time (88.2% medications, 77.6% glasses/contacts). Over 52.9% of respondents do not need dentures or hearing aids. Of those that do need these items, approximately 60% reported they could afford them always or most of the time (62.3% dentures, 59.1% hearing aids). See Figure 11.

## 4. Discussion

Over many years, the planning process of the Area Agency on Aging was the major source of information regarding service needs of older adults. This was an appropriate focus on individuals with greatest needs. In fact, a limitation of this study is that those most frail and living alone may have been the least likely to respond to the survey and may be under-represented. This is limitation is supported by the combined 4% reported to be living in supplemented housing and institutionalized care when we typically expect around 5% in US cities. We also see a comfortingly low level of reported frequent loneliness, although, again, this may be underreported.

However, the focus in not on those in greatest need, but to gather an assessment of the community as a whole from the perspective of the older members of the community. The Age-friendly City approach to looking at a community focusing on the eight domains gives the first multidimensional look at citizens over 50 and their self-reported assessment of the city. This is the first time that the city has been looked at in depth in terms of perceptions of positives and negatives by people ≥50. The very process of designing and carrying out the survey and the analyses has provided important information for focusing on priorities and possible intervention strategies. This information was collected pre-COVID-19. To take recent experiences into account, focus groups will be conducted in each ward of the city to determine what additional issues need to be considered in planning and priority setting.

One of the key findings was that older adults feel they need to have greater involvement in decision making and their opinions need to be considered by community leadership [13]. This is important as individuals make the transition from work to retirement as evidenced by the survey illustrating a sizeable group participating in work at some level [14]. It also speaks to why the assessment process as a precursor to priority setting is so important. Taking the time to fully explore the results, to understand the heterogeneity of views, and only then engage committees to discuss priorities in each of the eight domains is an important part of the prescribed process [6].

Another key finding is that staying in one’s home is a very high priority for Akron residents, higher than the national average for this item, typically 75% in the U.S. [15]. This also leads to a focus on the quality of the neighborhood on many dimensions such as safety, access to grocery shopping, access to health care, and the type of supports such as transportation and home services that are available [13]. Another limitation of the study is that it is apparent that not all areas of the city provide the same experience, and it is possible that those with the least positive experiences were less motivated to respond. However, there are sufficient indications that understanding how to intervein in selected areas with higher dissatisfaction rates will need to be determined.

Most important are the identification of gaps in needed services, need for alternative housing options, and the importance of having access to needed information for services when they become needed [16]. There are disparities that need to be focused on such as key sources of information for needed services. The results show that relatively few individuals were accessing the most helpful information sites and services. However, individuals felt that they were able to find information and felt that they were proactive when it is well known that individuals do not know what they need and only really look for information when there is a problem.

Another important area that was assessed is the quality of the living experience in terms of parks and recreations opportunities, opportunities to attend theater and concerts, attend sports events, engage in educational activities, and ability to have access to jobs and volunteer activities [17]. It becomes very important to carefully explore the areas that need to be addressed first.

To further support older adults in Akron aging in place, several community resources guides have been compiled, notably: the Senior Citizen Information Booklet, produced in Summit County and available online [18]; the Summit County Resource Guide, developed by Getting Wiser and Summit County 2-1-1 [19]; and the Akron Resource Directory, and online resources such as the 2-1-1 Summit County Resource Database.

Many agencies and resources referenced in these guides have found innovative ways to continue their work supporting older adults despite the COVID-19 pandemic. Programs offering minor home repairs at no cost such as Rebuilding Together of Northeast Ohio and Lift Up Ministries continue to provide valuable assistance with repairs around the home including roof/gutters, plumbing, electrical, porch stairs, and furnace. Although many adult day services have been temporarily suspended, organizations such as the Destination Home Akron Area Agency on Aging and Disabilities have been making telephonic wellness calls to residents they support. A plethora of meals programs, albeit with increased safety precautions, have continued their work delivering food to Akronites. In particular, Mobile Meals, Vantage Meals on Wheels, and others have been maintaining vital efforts in meal provision and delivery. As the city pivots to the future, it will be leveraging its strong, growth-ready programmatic infrastructure on which to build.

## 5. Conclusions

These results represent a summary of the first comprehensive assessment of the key domains of the City of Akron, a typical mid-sized American city, but unique in its history of efforts to provide comprehensive and wide-ranging services to older persons. The reported results document the rationale, tools, and comprehensive nature of the assessments of the Age-friendly cities and communities’ domains. They also demonstrate the degree of detail assessed to capture the accessibility and support for functional aging that is the hallmark of the program’s current approach.

The commitment to the Age-friendly cities and communities program represents a significant pivot in public policy for the City of Akron and Summit County, Ohio because it demonstrates a return to a unified effort of city government with regional aging services. This is particularly relevant because the City of Akron has a significant history of highly progressive aging services in its past, but a disjointed policy execution over the past two decades.

The analysis provides a comprehensive overview of the current state of each of the domains. It lays the groundwork for a discussion about community priorities. It allows a discussion that not only focuses on the requirements of those with the most need, which is of great importance, but also provides the opportunity to apply resources to improving the livability of the city. Thus, as the population continues growing older, more older adults are positioned to live independently and with greater life satisfaction.

As a summary of the current state of the domains in a typical mid-sized city, the results also provide a point of reference so that future assessments of other cities can make a comparison. Future research should be able to utilize studies such as these to test the accuracy of livability indices that are utilized to compare municipalities throughout the world.

## Figures and Tables

**Figure 1 ijerph-17-09103-f001:**
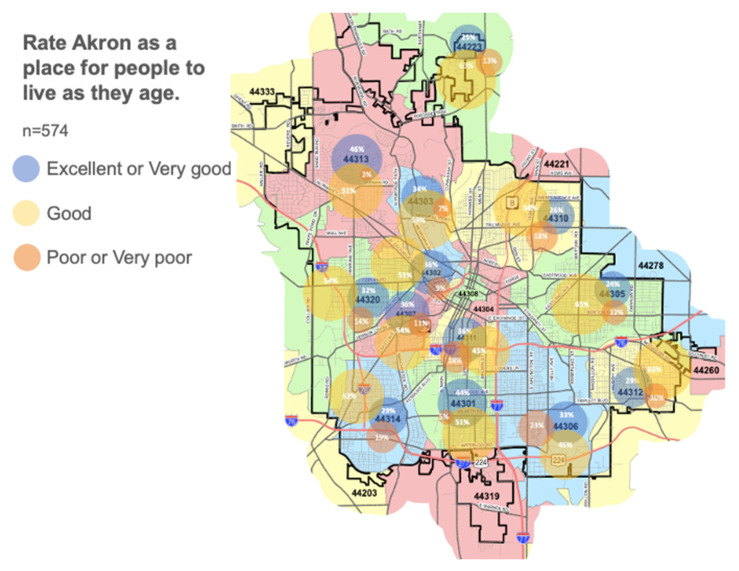
Ratings of the city of Akron as a place to live based on mailing zip codes.

**Figure 2 ijerph-17-09103-f002:**
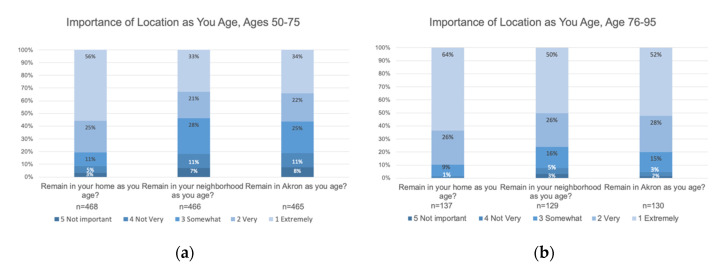
Respondents want most to remain in their homes, but they also want to remain in their neighborhoods and in the city of Akron: (**a**) the reported importance of aging in place for respondents aged 50–75; and (**b**) the reported importance of aging in place for respondents aged 76–95.

**Figure 3 ijerph-17-09103-f003:**
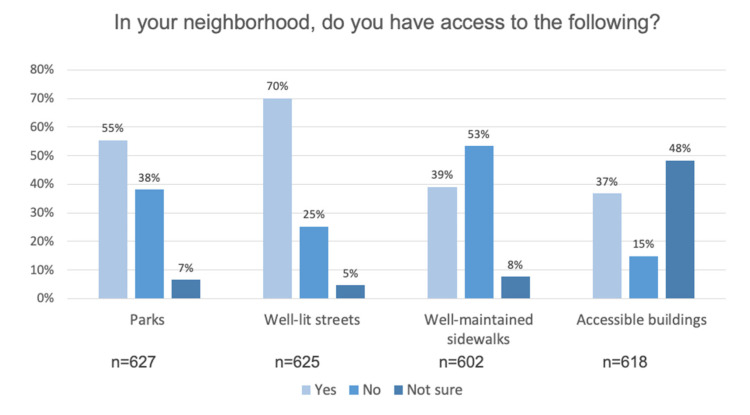
A majority of respondents indicated they have access to parks, well-lit streets, and accessible buildings. In contrast, access to well-maintained sidewalks was a concern.

**Figure 4 ijerph-17-09103-f004:**
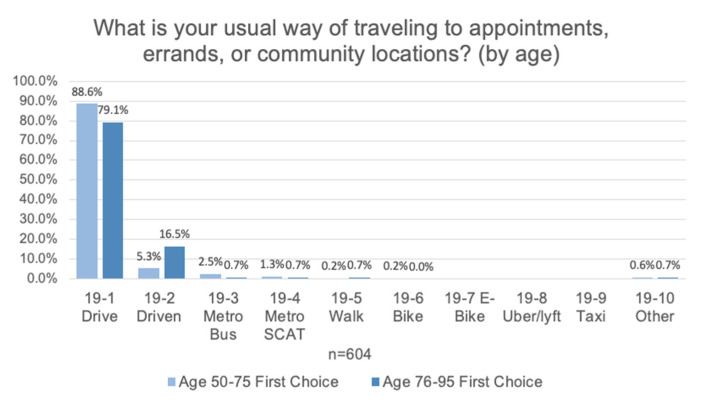
Respondents indicated that a vast majority of travel is done by driving themselves or being driven in a car. This is a low usage of standard public bus (Metro) and public on-demand services (SCAT) as well as walking.

**Figure 5 ijerph-17-09103-f005:**
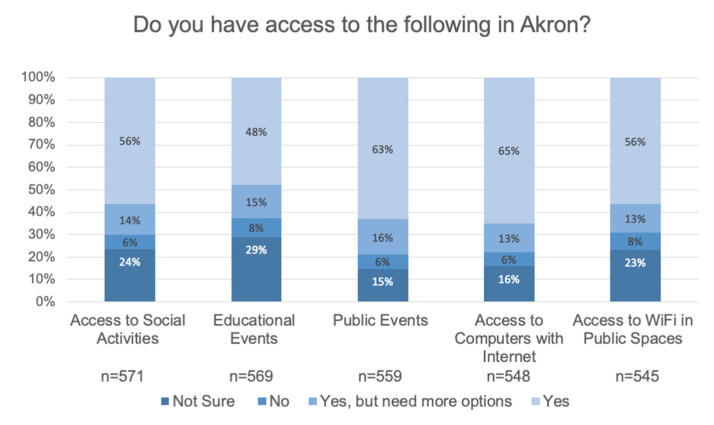
A summary of access social, educational and public events as well as access to computers and WiFi networks.

**Figure 6 ijerph-17-09103-f006:**
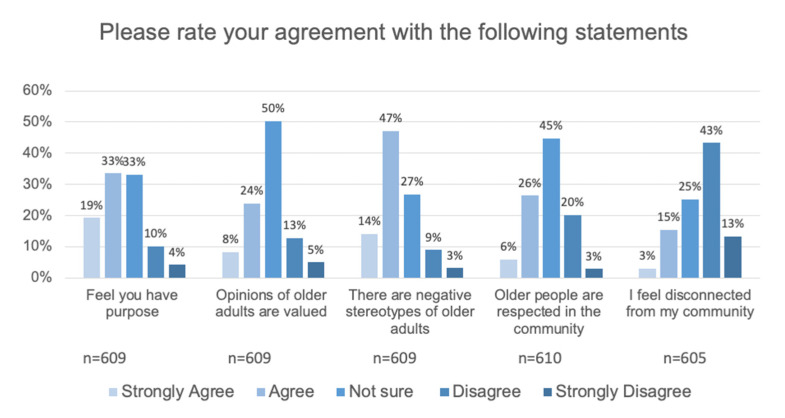
A majority of older adults report having purpose, while a minority indicate their opinions are valued and they are respected in the community. Encountering negative stereotypes of older adults as well as feeling disconnected are reported by a significant minority of respondents.

**Figure 7 ijerph-17-09103-f007:**
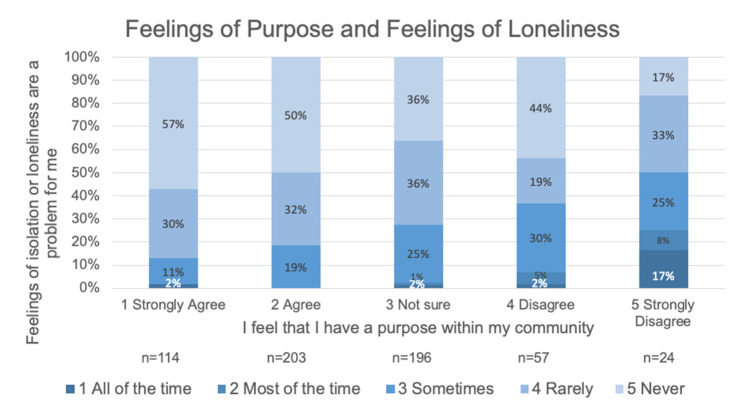
Feelings of purpose and loneliness of negatively related with greater purpose and less loneliness being the most reported state.

**Figure 8 ijerph-17-09103-f008:**
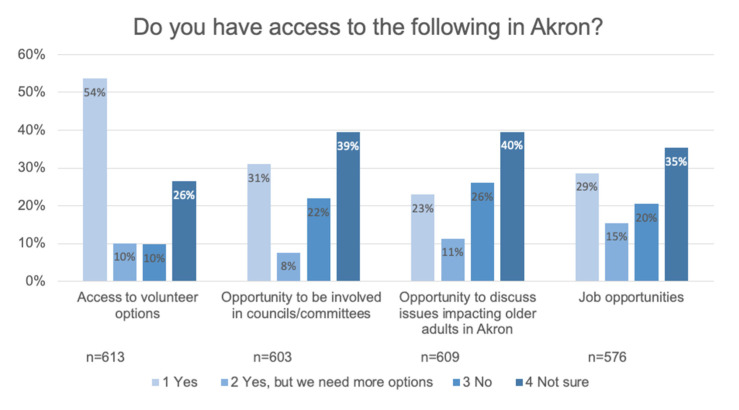
A majority of respondents had access to volunteer opportunities and opportunities to be involved in committees. Many respondents are not sure about various opportunities, which is an opportunity for improving awareness.

**Figure 9 ijerph-17-09103-f009:**
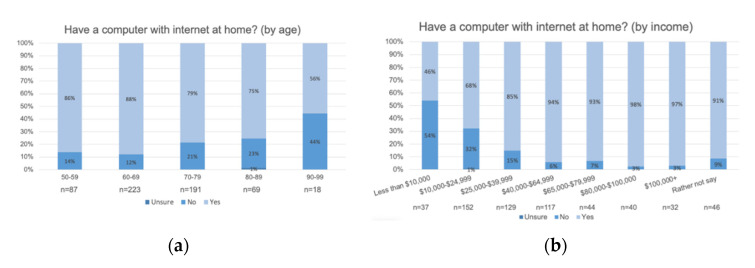
Respondents generally have computers with Internet in the city of Akron. Having a computer at home with Internet is: (**a**) negatively correlated with age (r_s_[588] = −0.135, *p* < 0.001); and (**b**) positively correlated with income (r_s_[597] = 0.329, *p* < 0.001).

**Figure 10 ijerph-17-09103-f010:**
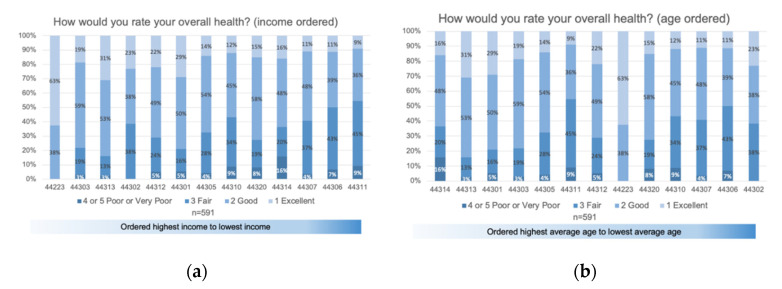
Respondents rate their overall current health status: (**a**) represents overall health ordered by zip code from highest to lowest incomes of the survey respondents; and (**b**) represents overall health ordered by highest average age to lowest average age of respondents. Zip codes with fewer than eight responses were excluded from this analysis.

**Figure 11 ijerph-17-09103-f011:**
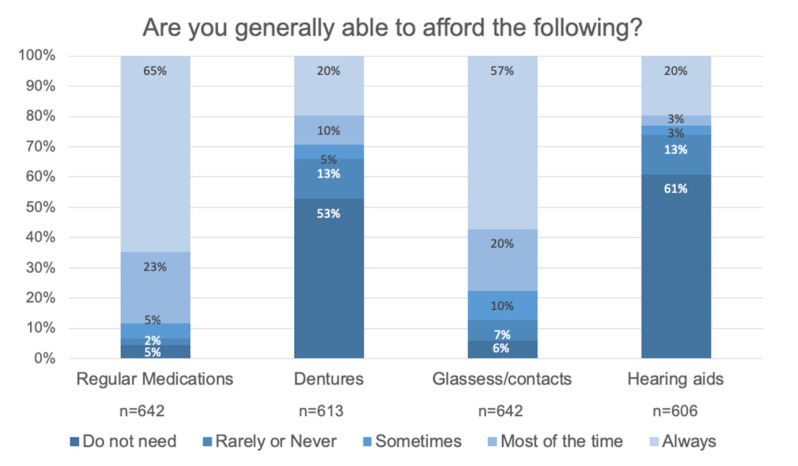
Affordability of regular medications, dentures, glasses/contacts, and hearing aids is reflected in this figure. The majority of respondents can afford regular medications and glasses/contacts always or most of the time. Around 60% of respondents do not need dentures (52.9%) or hearing aids (60.9%) currently.

**Table 1 ijerph-17-09103-t001:** Internal consistency measures of each survey domain.

Domain	Valid *n*	Cronbach’s α
Housing	568	0.672
Outdoor spaces	370	0.685
Transportation	175	0.804
Arts Entertainment and Leisure	627	0.838
Respect and Social Inclusion	569	0.692
Civic participation and employment	562	0.756
Communication	577	0.723
Health	291	0.690

**Table 2 ijerph-17-09103-t002:** Ratings of the city of Akron as a place to live based ordered by average income per mailing zip codes. Zip codes are ordered highest income to lowest income and those with less than 8 responses were excluded from this analysis.

	44223	44303	44313	44302	44312	44301	44305	44310	44320	44314	44307	44306	44311	Total
Excellent or Very good	25%	34%	46%	36%	29%	44%	24%	26%	32%	29%	36%	33%	36%	34%
Good	63%	59%	51%	55%	60%	51%	65%	56%	54%	52%	54%	45%	45%	55%
Poor or Very poor	13%	7%	3%	9%	10%	5%	12%	18%	14%	19%	11%	23%	18%	11%
n size	8	29	105	11	58	39	68	57	72	48	28	40	11	574

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
