# Peer review of "Prioritizing Age-Friendly Domains for Transforming a Mid-Sized American City"

_ijerph, 2020, doi:10.3390/ijerph17239103_

Round 1
Reviewer 1 Report
I do like the approach of your paper. I was happy to read about the return of interest for the elderly in Akron. I have some issues thou with your paper. I think you should try to position your work in a context of what other have done in the network of the Age Friendly Cities. I also think you should write more about the state of the art in this field of research. (It is often very helpful for the reader to find new references or to see that you have read the most important ones yourself.) As it stands now you give a very interesting picture about the development of the city’s activities for elderly.
Only 21.9% of the selected 3000 individuals did respond. This is unfortunately a very low degree of response and therefor you should discuss the implications of this. Who is not responding? Did you try to reach some of them to get a picture of the “non-responder”? When you say that loneliness is not an issue in Akron, I was wondering if the people who didn’t respond are the ones who are lonely. It mustn’t be like that but as a reader you don’t know. So please include a discussion about the low respond rate and what it could mean for the results.
The results are presented as a whole lot of tables and figures. I think it is to much, put some of it in the appendix. Tell us what the meaning of the figures are. I belong to those who think an article should tell me a story.
When you use number like 82.44% have a computer (p11, line 251) I can do some math and see that this is 515.25 individuals (as n=625 right?). With the number 82.4% this would have been 515 individuals. This makes me wonder if the n-number of 625 is wrong? Anyhow all this exercise with 1/100 of per cent gives a misleading impression of a precision which, I would say you don’t have. Sometimes you just give a number like 25% without any decimal at all. I think you should keep it simple and round it up to numbers with just one decimal, and then use it for all numbers even if it is just zero, i.e. 25% should be 25.0%.
When I look at your questions (thank you for providing them!) there are several I think have given you an interesting picture of what people in your selection think. I would have liked to read more about who they are, in what kind of housing they live, what kind of housing type they would consider moving to, and so on. I think you have much more information that could deepening the story of the elderly in Akron.
Reviewer 2 Report
- The introduction should explain the object of study and make reference to previous studies or questionnaires that analyze similar content. The need for this research must be justified, explaining the objectives and hypotheses pursued by the researchers.
- What were the criteria for inclusion and exclusion? It should be explained in the method
- Is the questionnaire previously validated? o If it has been created, has it gone through a validation process?
- How has the validation process of the questionnaire been? You must explain if it was done: face validity?, pilot study? clean collected data?, use principal components analysis (PCA)?, check de internal consistency?, are you revise your survey and edit final version?
- how reliable and valid is the measuring instrument? must it present these statistical data.
- Please explain the construct validity and concurrent validity of the questionnaire
- The study has passed through an Ethics Committee of the university, must indicate which and indicate the registration number. You must also indicate whether the study was conducted in accordance with the Declaration of Helsinki
- The method should describe what type of study was conducted. I understand that it is a cross-sectional study.
- I suggest that you present the descriptive statistics of the sample in a first table.
- Before results should explain the data analysis section.
- After that, statistical analyses should be performed again, justifying what kind of tests are performed for this study.
- Please include the limitations of the study.
- The discussion should be based on previous studies and only one reference is cited.
- A scientific study of this type cannot contain 9 references.
In addition, the questionnaire does not measure aspects as important as whether the person is physically active, what amount of physical activity is so important for the person's overall health and quality of life. Sports services are very important in the elderly and there is high evidence of savings in medical services
I recommend that you consult the recommendations of the American College of Sports Medicine and the Center for Disease Control.
This is a suggestion in case you want to take it into account to enrich the quality of your survey.
The work submitted has many shortcomings and could be rejected for publication but if you are considering doing so, I would be happy to review it again and support its publication.
Round 2
Reviewer 1 Report
Dear all, I appreciate the changes and corrections. You have now given us a better view of the situation in Arkon for the elderly by the material you have added.
I wish you good luck with further research in this area!
Reviewer 2 Report
The authors have made most of the changes requested so the article can be published